# ONLINE AGGLOMERATIVE POOLING FOR SCALABLE SELF-SUPERVISED UNIVERSAL SEGMENTATION

## ABSTRACT

Recent self-supervised image segmentors have achieved promising zero-shot performance. However, their pretraining schedule is multi-stage and alternates between offline pseudo-masks generation and parameters update, which leads to unstable training and sub-optimal solution. To solve this issue, we present Online Agglomerative Pooling (OAP) that allows efficiently generating universal pseudo-masks and updating parameters *simultaneously* at each training step. Specifically, OAP contains a stack of instance pooling and semantic pooling layers. By using a layer-varied threshold, OAP can generate multi-hierarchy masks that can provide more visual details for segmentation. Compared with MaskCut or Divide-Conquer, each OAP layer can identify connected nodes in parallel, thus can generate universal pseudo-masks for a single image within *tens of milliseconds*. Moreover, to deploy OAP in online pretraining, we devise a teacher-student framework with Query-wise Self-distillation, where the local view queries are each aligned with the matched global view queries to learn the local-to-global correspondence. Compared with other multi-stage offline pretraining methods, our framework can effectively scale to larger datasets while ensuring quicker convergence. Extensive experiments on the COCO, PASCAL VOC, Cityscapes, and UVO datasets show that our method achieves state-of-the-art performance on zero-shot instance segmentation, semantic segmentation, and panoptic segmentation. Our code and pretrained models shall be released upon acceptance of this work.

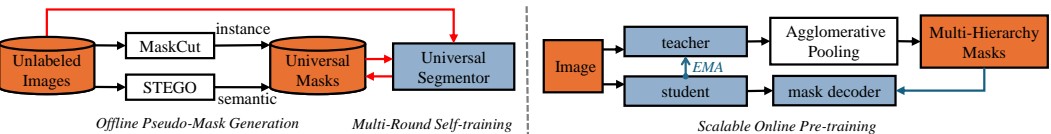

Figure 1: **Previous paradigm vs. our paradigm.** (left) Starting with merged universal pseudo-masks generated by different methods (Wang et al. (2023a; 2024); Hamilton et al. (2022)), previous paradigm (Niu et al. (2024); Wang et al. (2023a; 2024)) is multi-stage and alternates between updating parameters and offline re-generating new pseudo-masks. (right) We propose Online Agglomerative Pooling (OAP) that enables simultaneously generating masks efficiently and updating parameters in each training step. Compared to previous paradigm, our online framework effectively scales to large dataset and yields substantial performance improvements.

## 1 INTRODUCTION

Recent years have witnessed significant advancements in image segmentation (Xie et al. (2021); Cheng et al. (2022); Kirillov et al. (2023); Ravi et al. (2024)). However, these models typically require extensive human annotations for training, which is exceptionally time-consuming and labor-intensive; for example, annotating a single image in the SA-1B (Kirillov et al. (2023)) dataset can take over 25 minutes of detailed work. Moreover, human annotations often contain noise and inconsistencies, making them sometimes poorly aligned with the fine details of the images. Additionally, human annotations are susceptible to personal biases, as different annotators may have varying interpretations of what constitutes an instance or how semantic categories are defined. These inherent

Table 1: Comparison of pseudo-mask generation time and performance among different algorithms. We run all methods on COCOval2017, evaluate their average processing time per image (in seconds) and average zero-shot class-agnostic instance segmentation performance. All images are scaled to a resolution of 448. All methods use the DINO pretrained ViT-base/8 backbone to extract the features.

| method | time | $AP^{mask}$ |
|---|---|---|
| TokenCut (Wang et al. (2023b)) | 2.25 | 3.5 |
| MaskCut (Wang et al. (2023a)) | 4.72 | 6.7 |
| Divide-and-Conquer (Wang et al. (2024)) | 5.27 | 7.6 |
| Online Agglomerative Pooling | 0.045 | 6.2 |

shortcomings in manual annotations can compromise the robustness and generalizability of segmentation models, potentially limiting their efficacy across various visual contexts. In this paper, we aim to explore the design of a self-supervised pretraining model for image segmentation that leverages the intrinsic information within the images themselves, reducing reliance on extensive human-labeled data.

To reduce reliance on manual annotations and learn from visual data themselves, several self-supervised zero-shot instance segmentors (Wang et al. (2023a; 2024); Niu et al. (2024); Arica et al. (2024)) are proposed. They first use pretrained SSL features and graph-partitioning (Wang et al. (2023b;a) algorithms to generate pseudo-masks for each image. As shown in Table 1, although these offline algorithm has better zero-shot performance for initialized training, they are very time-consuming. They can not be deployed to online environment and the offline pseudo-masks generation is also very time-consuming. Most importantly, the pseudo-masks used for the next round training is directly extracted from the last checkpoint, which makes the training unstable when switching to another round. This may lead to sub-optimal solution.

The key solution for enabling online pretraining is to reduce the processing time of pseudo-mask generation algorithm. To achieve this goal, we propose the **Online Agglomerative Pooling (OAP)** algorithm to efficiently generate *universal* pseudo-masks of one image within tens of milliseconds. Our key insight is that semantically-similar patches are spatially close *in group*. We can identify these groups of strongly-connected nodes *in parallel* (Pearce (2005)). As shown in Table. 1, OAP can generate comparable zero-shot instance segmentation performance to previous methods but is $100\times$ faster than previous methods, which enables online large-scale pretraining for self-supervised universal segmentation.

Equipped with the much more efficient pseudo-masks generation algorithm, we propose to adopt a teacher-student framework with **Query-wise Self-distillation** as the pretext task to train the self-suerpervised segmentation model. Unlike previous self-supervised representation models that utilize global pooling or the ViT [CLS] (Dosovitskiy (2020)) token, we condense each image into a set of universal queries, and train each student query to predict the corresponding bipartite-matched teacher query. We empirically find that this simple but effective approach benefits self-supervised segmentation by enabling more fine-grained learning of individual segments instead of focusing just on the global context.

Our main contributions are:

- We propose an efficient pseudo-masks generation algorithm, Online Agglomerative Pooling (OAP), which enables to generate high-quality semantic-level and instance-level pseudo-masks within tens of milliseconds. This enables large-scale online pretraining.

- We propose the Query-wise Self-distillation loss to pretrain universal segmentation models in a self-supervised manner. As far as we know, our model is the first work for online self-supervised universal segmentation. Compared with other multi-stage alternating frameworks, our model converges faster and achieves significant performance improvements.

- Extensive experiment results on COCO, PASCAL VOC, UVO, and Cityscapes validate that our model achieves state-of-the-art performance on zero-shot self-supervised instance segmentation, semantic segmentation and panoptic segmentation.

## 2 RELATED WORK

**Self-supervised Representation Learning.** Self-supervised representation learning aims to learn universal (Bengio et al. (2013) features from large amounts of unlabeled instances without manual annotation. A pretext task is often pre-defined to train the model. According to types of pretext task, they can be classified into *contrastive learning* methods and *masked image modelling* methods. Contrastive learning based methods include pretexts based on negative samples (Chen et al. (2020a); He et al. (2020); Chen et al. (2021)), clustering (Caron et al. (2020); Asano et al. (2019)), self-distillation (Caron et al. (2021); Grill et al. (2020); Chen & He (2021)), feature decorrelation (Zbontar et al. (2021); Bardes et al. (2021)). Masked image modelling methods include pretexts tasks based on low-leve targets (He et al. (2022); Chen et al. (2020b); Wei et al. (2022)), high-level targets (Bao et al. (2021); Dong et al. (2023); Chen et al. (2024), self-distillation (Chen et al. (2022a); Baevski et al. (2022)), and multi-modal teacher (Zhou et al. (2022); Peng et al. (2022)).

**Self-supervised Instance and Semantic Segmentation.** Recently studies (Caron et al. (2021); Hamilton et al. (2022); Siméoni et al. (2021); Vo et al. (2020; 2021)) show pretrained SSL features can capture pixel-to-pixel semantic similarity. Inspired by that, several works (Hamilton et al. (2022); Wang et al. (2023a); Arica et al. (2024); Wang et al. (2024); Seong et al. (2023); Kim et al. (2024); Wang et al. (2023b); Van Gansbeke et al. (2022); Wang et al. (2022); Liu et al. (2024)) aim to distill or self-train a segmentation model based on the pretrained SSL representation. These methods can be classified into semantic segmentation methods, zero-shot instance segmentation methods, and universal segmentation methods. State-of-the-art unsupervised zero-shot instance segmentation methods (Wang et al. (2023a); Arica et al. (2024); Wang et al. (2024) adopt a cut and learn pipeline, in the sense that they first generate pseudo-masks using pretrained SSL representation, and then learn a model through multi-round self-training. Unsupervised semantic segmentation methods (Hamilton et al. (2022); Seong et al. (2023); Kim et al. (2024); Liu et al. (2024) adopts a distillation-based objective, in the sense that the projected segmentation features should preserve the pixel-to-pixel semantic correspondence in the original SSL representation space. Recently, U2Seg (Niu et al. (2024)) proposes a self-supervised universal segmentation framework for semantic, instance, and panoptic segmentation. They adopt the similar cut-and-learn pretraining pipeline. To make the model semantic-aware, they cluster the masks to generate the semantic pseudo-labels for semantic training.

In contrast, we propose Online Agglomerative Pooling, which is an efficient pseudo-mask generation algorithm, that enables pretraining self-supervised segmentation models in an online manner and effectively scaling to large datasets, without any offline generation or clustering. As far as we know, our work is the first online method for self-supervised instance and universal segmentation.

**Graph Pooling.** As an esential component of Graph Neural Networks, graph pooling is important to obtain a holistic graph-level representation. Graph pooling can be roughly divided into *flat pooling* and *hierarchical pooling* according to its role in graph-level representation learning. Flat pooling (Dwivedi et al. (2023); Xu et al. (2020); Noutahi et al. (2019)), also known as Graph Readout, aims to obtain a global graph-level representation. Hierachical pooling aims to iteratively coarsening the graph into smaller size. It can be classified into node clustering pooling (Wu et al. (2019); Liu et al. (2021)) and node drop pooling (Gao et al. (2019); Lee et al. (2021); Gao et al. (2021)).

## 3 PRELIMINARIES

**Unsupervised Universal Image Segmentation (U2Seg).** Universal segmentation requires annotations of instance-level masks, semantic-level masks, and the class label for each mask. As the first self-supervised universal segmentor, U2Seg (Niu et al. (2024)) follows a similar cut-and-learn (Wang et al. (2023a)) pretraining pipeline. The "cut" stage is offline and generates the pseudo universal annotations for the whole dataset. Specifically, based on the self-supervised feature maps (Caron et al. (2021)), U2Seg uses the graph-partitioning algorithm MaskCut (Wang et al. (2023a)) to generate a set of instance-level pseudo-masks for each image. K-means is then used to cluster these masks and get their pseudo-labels. For semantic-level annotations, U2Seg uses a distillation-based model STEGO (Hamilton et al. (2022)) to get the pixel-wise semantic pseudo-label. The "learn" stage alternates between training model using the current annotations and generating new pseudo universal annotations using the previous model checkpoint.

**Self-distillation with No Labels (DINO).** Self-supervised representation learning frequently incorporates a pretext task to supervised encoder training. DINO (Caron et al. (2021)) is a self-distillation model that leverages online clustering (Caron et al. (2020); Asano et al. (2019). Specifically, DINO employs a student-teacher framework wherein both networks consist of an encoder and a projection head designed for online clustering. The teacher network encodes solely global views of the image, whereas the student network processes both global and local views created through multi-crop augmentation (Caron et al. (2020). The student head learns by aligning its outputs to those of the teacher head:

$$SD(\mathbf{p}_t, \mathbf{p}_s) = -\sum_{k=1}^{K} \mathbf{p}_t^k \log \mathbf{p_s}^k, \tag{1}$$

where $K$ is the online cluster number, $\mathbf{p}_t \in R^K$ is the softmax code of the teacher view, $\mathbf{p}_s \in R^K$ is the softmax code of the student view. The self-distillation loss between local and global view enables the model to capture local-to-global correspondences. The teacher network is updated using a momentum mechanism (He et al. (2020)) that effectively ensembles the model over time (Tarvainen & Valpola (2017)), thus providing higher-quality target features to guide the student and enhance learning without the need for manual labels.

## 4 SCALABLE SELF-SUPERVISED UNIVERSAL SEGMENTATION

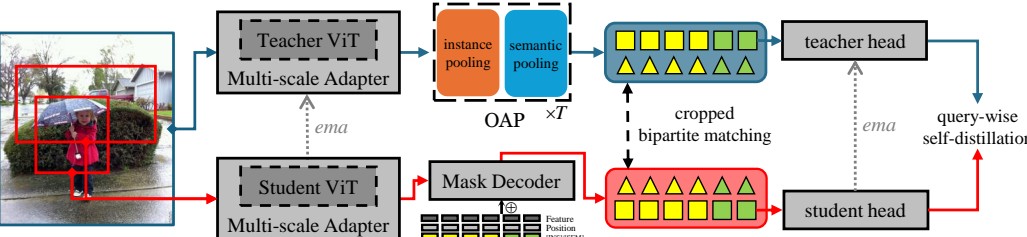

Figure 2: Our online pretraining framework for scalable self-supervised universal segmentation. We adopt a teacher-student framework. The original view (global) is fed into the teacher branch, then we propose to use Online Agglomerative Pooling (OAP) to efficiently generate semantic/instance-level masks ($\triangle$) and their feature embeddings ($\square$). In student branch, we use mask decoder with level-specific object queries to predict the local view universal masks. To train the network, we use bipartite matching to match the student outputs with the cropped teacher masks. We propose Query-wise Self-distillation to align each student query with corresponding matched teacher query.

### 4.1 OVERVIEW ARCHITECTURE

As shown in Figure 2, given an image, we use multi-crop (Caron et al. (2020)) augmentation to get a set of local views from the original view. The original view is input to the teacher branch, where a multi-scale encoder (Chen et al. (2022b); Caron et al. (2021) extracts its features, and a series of Online Agglomerative Pooling (OAP) layers are then employed to generate the instance-level and semantic-level pseudo-masks along with their feature embeddings. Simultaneously, each local view is fed to the student encoder to first extract its features. A mask decoder (Cheng et al. (2022)) is then used to predict its universal masks and feature embeddings. We partition the object queries into two distinct parts—semantic queries, distinguished by the learnable token [SEM], which focus on capturing semantic-level information across the image, and instance queries, distinguished by the learnable token [INS], which aim to identify individual instances within an image.

After both branches have generated their respective universal masks and feature embeddings, we treat teacher outputs as targets and perform bipartite matching to align the student outputs. Since the teacher processes the global view, we spatially crop the teacher's pseudo-masks to correspond with the student's local views before matching.

We also incorporate the pretext task, online clustering with self-distillation (Caron et al. (2020; 2021)), to train the student branch. Unlike DINO or SwAV, which distills a single global image fea-

ture, query-wise self-distillation uses multiple query features per image corresponding to different segments. After matching, each student query learns from its teacher counterpart. This approach benefits self-supervised segmentation by enabling fine-grained learning of individual segments.

Unlike previous seminal works Wang et al. (2023a; 2024); Niu et al. (2024); Arica et al. (2024), our pretraining framework does not include any offline pseudo-masks generation or offline clustering, thereby enabling self-supervised segmentation models to scale effectively to larger datasets. Moreover, as shown in Section 5, continuously updating pseudo-annotations allows our model to converge more rapidly and reliably, avoiding the loss fluctuations associated with previous alternating multi-stage frameworks.

## 4.2 ONLINE AGGLOMERATIVE POOLING

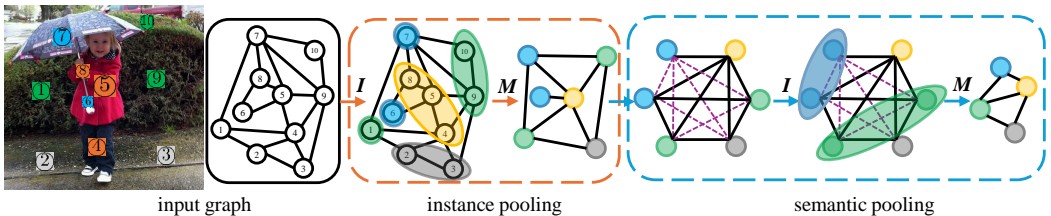

input graph        instance pooling        semantic pooling

Figure 3: One layer in Online Agglomerative Pooling (OAP). Each layer first generate instance-level pseudo-masks using instance pooling. Based on the result of instance pooling, a fully-connected graph is built. Semantic-level pseudo-masks are then generated by semantic pooling. The result of instance pooling is used as input in the next OAP layer. Both instance pooling and semantic pooling adopt the *I*dentify then *M*erge pipeline.

As shown in Figure.3, each OAP layer first finds groups of strongly-connected nodes, and merge each group into one supernode. Compared with the optimization-based TokenCut (Wang et al. (2023b)) and MaskCut (Wang et al. (2023a)) methods, OAP is a heuristic approach and does not require computation of eigenvectors. Moreover, thanks to the SCC (Pearce (2005)) algorithm, OAP can group nodes in parallel, wich is also much faster than the Divide-and-Conquer approach adopted in (Wang et al. (2024)). Next we will illustrate the initialization, identify step, and the merge step in details. We summarize Online Agglomerative Pooling in Algorithm. 1.

**Graph Initialization.** Following (Wang et al. (2023a; 2024)), we use the L2-normalized "key" features $\mathbf{F} \in R^{H'W' \times d}$ from the last self-attention layer in the teacher encoder to initialize the graph. Specifically, each token is treated as one node, and edges are formed solely between nodes that are directly adjacent horizontally and vertically. Throughout the pooling process, each node $i$ is associated with a mask $\mathbf{M}_i \in \{0,1\}^{H'W'}$ denoting which tokens belong to its subtree. We initialize $\mathbf{M}^0$ as the identity matrix $\mathbf{I} \in \{0,1\}^{H'W' \times H'W'}$. The initialized *undirected, connected* graph is denoted as $\mathcal{G}^0 = \{\mathbf{V}^0, \mathbf{M}^0, \mathbf{E}^0\}$, where $\mathbf{V}^0 \in R^{s^0 \times d}$, $\mathbf{E}^0 \in \{0,1\}^{s^0 \times s^0}$ is the adjacency matrix, $s^0 = H'W'$ is nodes number. We also denote $\mathbf{A} = \mathbf{F}\mathbf{F}^T \in R^{H'W' \times H'W'}$ as the spatial affinity matrix, which are used when computing the edge similarity.

**Identify Step.** For semantically-similar tokens, their features should have large cosine similarity, therefore, we compute the *feature similarity* of two adjacent nodes as:

$$\mathcal{S}_{ij}^f = \mathbf{V}_i^t \mathbf{V}_j^{t\,T}. \tag{2}$$

However, due to the merge step (which will be illustrated later), node features are evolving after each layer. The pairwise similarity may not be consistent with the one implied from the original encoder features. To mitigate this issue, we also measure the *spatial similarity* of two nodes as:

$$\mathcal{S}_{ij}^s = 1 - \frac{1}{H'W'} |\frac{\mathbf{M}_i^t \mathbf{A}}{\mathbf{M}_i^t \mathbf{1}^T} - \frac{\mathbf{M}_j^t \mathbf{A}}{\mathbf{M}_j^t \mathbf{1}^T}| \mathbf{1}^T, \tag{3}$$

where $|\cdot|$ is the absolute operator, $\mathbf{1} \in R^{1 \times H'W'}$ is a vector of 1s. node $i$ and $j$ are considered semantically similar if they have similar affinity distribution along the original tokens map. $\mathcal{S}_{ij}^s$

does not require direct feature comparison, it assembles a voting mechanism, in the sense that each original token gives its score on their similarity. The final similarity measure is formulated as:

$$\mathcal{S}_{ij} = \omega_f \mathcal{S}_{ij}^f + \omega_s(\mathcal{S}_{ij}^s), \tag{4}$$

where $\omega_f + \omega_s = 1$ and $\mathcal{S}_{ij} \in [-1, 1]$. Then, given a threshold $\tau_t$, edges with similarity measure larger than $\tau_t$ are labeled to be contracted. We use the SCC (Pearce (2005); Scipy (2024)) algorithm to find a set of connected groups of nodes. Nodes in each group will be merged into one supernode.

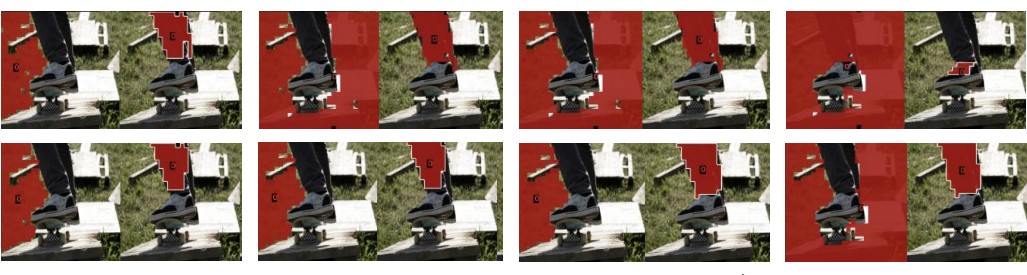

coarsen

Figure 4: Mask visualization of different feature updating strategy. Each column corresponds to one OAP layer ($t$=2,3,4,5). (Up) The supernode feature is updated using $\mathbf{V}_i^{t+1} = \mathbf{\Omega}^T \mathbf{V}_i^t$. (Bottom) The supernode feature is updated using Equation. 6

**Merge Step.** The SCC algorithm outputs a node-supernode assignment matrix $\mathbf{\Omega} \in \{0,1\}^{s^t \times s^{t+1}}$. According to the assignment matrix, the new adjacent matrix and mask matrix is updated as:

$$\mathbf{E}^{t+1} = \mathbf{\Omega}^T \mathbf{E^t}(\mathbf{\Omega}^T \mathbf{E^t})^T; \mathbf{M}^{t+1} = \mathbf{\Omega}^T \mathbf{M}^t, \tag{5}$$

where each supernode takes a union of each child node mask. To get the supernode features, a direct solution is averaging features of tokens inside its mask, i.e. $\mathbf{V}_i^{t+1} = \mathbf{\Omega}^T \mathbf{V}_i^t$. However, we empirically find that this can cause semantically-different but spatially-close nodes to first merge. We think this is due to that the supernode feature includes information of its boundary tokens and is not representative of its main region. When two supernodes are neighboring to each other, their common boundary tokens will make their feature having a large similarity measure. To mitigate this issue, similar to Equation.3, the supernode feature is computed as:

$$\mathbf{V}_i^{t+1} = \text{L2N}\{\text{softmax}(\frac{\mathbf{M}_i^{t+1}\mathbf{A}}{\sigma})\mathbf{F}\}, \tag{6}$$

where L2N$\{\cdot\}$ denotes L2-normalization, $\sigma$ is the softmax temperature. Equation.6 can also be seen as a voting mechanism, where most of the softmax mass will lie on the salient region of the supernode, and the information of boundary tokens is suppressed. As shown in Figure. 4, the "grass" region are not merged with the "human leg" region throughout the pooling process. This shows that updating supernode features using Equation 6 can make semantically-different but spatialy-close regions more discriminative.

**Instance Pooling and Semantic Pooling.** It is noteworthy that edges exist only between adjacent nodes in $\mathcal{G}^0$. Moreover, the edge updating function Equation. 5 ensures that edges are maintained only between adjacent supernodes. Consequently, each mask corresponds to a single connected region, which is predominantly suitable for instance segmentation tasks. However, semantic segmentation often requires masks that encompass multiple disjoint regions belonging to the same semantic class. To address this limitation, we construct a fully connected version of the graph derived from instance pooling. Subsequently, the same Identify and Merge pipeline is employed to generate semantic masks, thereby accommodating multiple disjoint components within a single semantic mask. The graph derived from instance pooling is used as input of the next OAP layer.

**Time-varied Thresholding.** Online Agglomerative Pooling can be seen as a graph coarsen procedure, where the coarsened supernodes represent its whole receptive region and removes the abundant information among adjacent pixels. Therefore, we use a decreasing threshold to filter out masks at different semantic hierarchy. We empirically find that the background tokens, where pixels are very similar to their neighbors, are merged at the very early OAP layers by using a high threshold (such as 0.9). While tokens like "human head" and "human body" with higher level semantics are merged at later OAP layers using a lower threshold (such as 0.5).

---

**Algorithm 1** Online Agglomerative Pooling

---

**Require:** Initialized graph: $\mathcal{G}^0 = \{\mathbf{V}^0, \mathbf{M}^0, \mathbf{E}^0\}$, Spatial affinity matrix: $\mathbf{A} \in R^{H'W' \times H'W'}$,
L2-normalized key features: $\mathbf{F} \in R^{H'W' \times d}$, Layer-varied thresholds: $\{\tau_t\}_{t=1}^{T}$, Softmax temperature: $\sigma$, Spatial and Feature similarity weight: $\omega_s, \omega_f$, Mask area threshold $\phi$

---

1: *instance masks* ← {}, *instance features* ← {}, *semantic masks* ← {}, *semantic features* ← {}
2: **for** $t$ **in** $[0, ..., T-1]$ **do**
3:     Compute similarity score of each edge using Equation.4
4:     Find set of edges $\{E_i^t\}$ with similarity score larger than $\tau_t$
5:     Compute node-supernode assignment matrix $\mathbf{\Omega} = \text{SCC}(\{E_i^t\})$
6:     Update $\mathbf{E}^{t+1} \leftarrow \mathbf{\Omega}^T \mathbf{E}^{\mathbf{t}} (\mathbf{\Omega}^T \mathbf{E}^{\mathbf{t}})^T$
7:     Update $\mathbf{M}^{t+1} \leftarrow \mathbf{\Omega}^T \mathbf{M}^t$
8:     **for** $i$ **in** $[0, ..., s^{t+1}]$ **do**
9:         Update $\mathbf{V}_i^{t+1} \leftarrow \text{L2N}\{\text{softmax}(\frac{\mathbf{M}_i^{t+1} \mathbf{A}}{\sigma}) \mathbf{F}\}$
10:         **if** $area(\mathbf{M}_i^t) \geq \phi$ **then**
11:             Append $\mathbf{M}_i^t$ to *instance masks*, Append $\mathbf{V}_i^t$ to *instance features*
12:         **end if**
13:     **end for**
14:     Build a new fully-connected graph $\mathcal{G}^s = \{\mathbf{V}^{t+1}, \mathbf{M}^{t+1}, \mathbf{E}^s\}$
15:     Repeat steps 3-13 for $\mathcal{G}^s$, get *semantic features* and *semantic masks*.
16: **end for**
17: **return** *semantic masks, semantic features, instance masks, instance features*

---

## 4.3 MODEL TRAINING

To deploy Online Agglomerative Pooling (OAP) for scalable online self-supervised universal segmentation, we devise a teacher-student framework with momentum updating strategy. Moreoever, we introduce a novel pretext task, Query-wise Self-Distillation, specifically designed to train models for self-supervised segmentation.

**Multi-scale Encoder and Zero-initialization.** To utilize the important multi-scale information ( Lin et al. (2017); Cheng et al. (2022)) for segmentation tasks, existing self-supervised models ( Wang et al. (2023a; 2024); Niu et al. (2024)) utilize the DINO pretrained ResNet ( He et al. (2016)). To further benefit from large-scale ViT( Dosovitskiy (2020)) pre-training, we devise Vit-Adapter( Chen et al. (2022b)) as our multi-scale encoder. Given that OAP is a non-parametric module without prior information, we zero-initialize the adapter parameters so that OAP can generate meaningful masks during the early stages of training.

**Level-specific Object Queries.** To train a universal segmentation model, the mask decoder must output both instance-level masks and semantic-level masks. Unlike U2Seg (Niu et al. (2024)) where the decoder has two separate branches, we adopt a more streamlined approach by partitioning the object queries in the mask decoder into the semantic queries and instance queries. To distinguish between these two groups, we also introduce two special learnable tokens [INS] and [SEM], which are added to the respective query features at each decoder layer. It is noteworthy that both groups of queries share the same decoder parameters and projection head, enabling universal learning through parameter sharing across different levels.

**Cropped Bipartite Matching.** The teacher network processes the original *global* view of the image, producing pseudo-masks that cover the entire image. In contrast, the student network processes *local* views obtained from multi-crop augmentation. To resolve the mismatch, we crop the teacher's pseudo-masks to match the spatial regions of the student's local views. Teacher embeddings and pseudo-masks that does not overlap with the student local view are dropped. We employ the Dice similarity coefficient as the matching criterion. Moreover, the matching is conducted separately within each segmentation level, in the sense that the matching processes for the semantic and instance levels are independent and do not interfere with each other.

**Query-wise Self-Distillation.** We denote the query embeddings of the teacher and student branch after bipartite matching as $\mathbf{Q}_t \in R^{L \times d}$ and $\mathbf{Q}_s \in R^{L \times d}$, where $L$ is the query number. We propose

a simple but effective loss specifically designed for self-distilled segmentation:

$$\sum_{i=1}^{L} SD\{\text{teacher head}(\mathbf{Q}_t^i), \text{student head}(\mathbf{Q}_s^l)\}. \tag{7}$$

Loss. 7 is simply a sum of self-distillation loss over each pair of matched query embedding. This is different from the representation-specific DINO loss. 1 where the code is the projection of the global ViT `[CLS]` token. This approach benefits self-supervised segmentation by enabling more fine-grained learning of individual segments instead of just focusing on the global context.

# 5 EXPERIMENTS

Table 2: Performance comparison with previous methods on class-agnostic zero-shot instance segmentation, zero-shot instance-segmentation, unsupervised semantic segmentation, and panoptic segmentation. Our model outperforms other state-of-the-art performance on all tasks.

| Task → | Agn Instance Seg. | | Instance Seg. | | | | | | Semantic Seg. | | Panoptic Seg. | | | | | |
|---|---|---|---|---|---|---|---|---|---|---|---|---|---|---|---|---|
| Datasets → | COCO | | COCO | | VOC | | UVO | | COCO | | COCO | | | Cityscapes | | |
| Metric → | $AP_{50}^{box}$ | $AP^{box}$ | $AP_{50}^{box}$ | $AR_{100}^{box}$ | $AP_{50}^{box}$ | $AR_{100}^{box}$ | $AP_{50}^{box}$ | $AR_{100}^{box}$ | PixelAcc | mIoU | PQ | SQ | RQ | PQ | SQ | RQ |
| FreeSOLO Wang et al. (2022) | 9.6 | 4.2 | - | - | - | - | - | - | - | - | - | - | - | - | - | - |
| TokenCut Wang et al. (2023b) | 5.8 | 3.2 | - | - | - | - | - | - | - | - | - | - | - | - | - | - |
| CutLER Wang et al. (2023a) | 21.9 | 12.3 | - | - | - | - | - | - | - | - | - | - | - | - | - | - |
| DINO Caron et al. (2021) | - | - | - | - | - | - | - | - | 30.5 | 9.6 | - | - | - | - | - | - |
| STEGO Hamilton et al. (2022) | - | - | - | - | - | - | - | - | 56.9 | 28.2 | - | - | - | - | - | - |
| CutLER+ | - | - | 9.0 | 10.3 | 26.8 | 27.2 | 10.6 | 11.8 | - | - | - | - | - | - | - | - |
| CutLER+STEGO | - | - | - | - | - | - | - | - | - | - | 12.4 | 64.9 | 15.5 | 12.4 | 36.1 | 15.2 |
| U2Seg Niu et al. (2024) | 22.8 | 13.0 | 11.8 | 21.5 | 31.0 | 48.1 | 10.8 | 25.0 | 63.9 | 30.2 | 16.1 | 71.1 | 19.9 | 17.6 | 52.7 | 21.7 |
| Ours | 28.7 | 19.6 | 25.3 | 30.5 | 38.4 | 56.4 | 16.2 | 32.1 | 68.4 | 36.1 | 20.2 | 80.6 | 26.7 | 20.5 | 60.1 | 29.7 |

Table 3: Comparison with other methods on unsupervised object detection and instance segmentation on UVO`val` and COCO`val` and comparison with other methods on unsupervised panoptic segmentation on Cityscapes`val` and COCO`val`. Our model outperforms previous methods by a large marin on all univeral segmentation settings

| Metric (UVO val-Instance) | $AP^{box}$ | $AP_{50}^{box}$ | $AR_{100}^{box}$ | $AP^{mask}$ | $AP_{50}^{mask}$ | $AR_{100}^{mask}$ |
|---|---|---|---|---|---|---|
| CutLER+Wang et al. (2023a) | 6.3 | 10.6 | 11.8 | 6.0 | 9.0 | 10.4 |
| U2SegNiu et al. (2024) | 6.8 | 10.8 | 25.0 | 6.2 | 9.5 | 21.0 |
| Ours | 10.1 | 16.2 | 32.1 | 8.3 | 12.1 | 24.3 |

| Metric(COCOval-Instance) | $AP^{box}$ | $AP_{50}^{box}$ | $AR_{100}^{box}$ | $AP^{mask}$ | $AP_{50}^{mask}$ | $AR_{100}^{mask}$ |
|---|---|---|---|---|---|---|
| CutLER+Wang et al. (2023a) | 5.9 | 9.0 | 10.3 | 5.3 | 8.6 | 9.3 |
| U2SegNiu et al. (2024) | 7.3 | 11.8 | 21.5 | 11.2 | 6.4 | 18.5 |
| Ours | 9.6 | 25.3 | 30.5 | 14.6 | 12.4 | 23.3 |

| Methods (Cityscapesval-Panoptic) | Pretrain | PQ | SQ | RQ |
|---|---|---|---|---|
| *zero-shot methods* | | | | |
| U2SegNiu et al. (2024) | IN | 15.7 | 46.6 | 19.8 |
| Ours | IN | 20.1 | 53.4 | 24.3 |
| *non zero-shot methods* | | | | |
| CutLER+STEGO | COCO | 12.4 | 36.1 | 15.2 |
| U2SegNiu et al. (2024) | COCO+IN | 17.6 | 52.7 | 21.7 |
| Ours | COCO+IN | 20.5 | 60.1 | 29.7 |

| Methods(COCOval-Panoptic) | Pretrain | PQ | SQ | RQ |
|---|---|---|---|---|
| *zero-shot methods* | | | | |
| U2SegNiu et al. (2024) | IN | 11.1 | 60.1 | 13.7 |
| Ours | IN | 16.2 | 71.2 | 18.5 |
| *non zero-shot methods* | | | | |
| CutLER+STEGO | COCO | 12.4 | 64.9 | 15.5 |
| U2SegNiu et al. (2024) | COCO+IN | 16.1 | 71.1 | 19.9 |
| Ours | COCO+IN | 20.2 | 80.6 | 26.7 |

## 5.1 TRAINING SETTINGS

**Model Architecture.** For the multi-scale encoder, we use the DINO pretrained ViT-base/8 and ViT-Adapter(Chen et al. (2022b)). For the mask decoder, we use the official setting of Mask2Former(Cheng et al. (2022)). For the project head, we follow DINO(Caron et al. (2021)) using a 3-layer MLP with hidden dimension 2048 followed by L2 normalization and a linear layer of $K$ dimensions. The adapter parameters are zero-initialized for stable pretriaining. After training, the final teacher network with the mask decoder is used for inference. For unsupervised semantic segmentation, we follow (Niu et al. (2024); Hamilton et al. (2022)) to additionaly fine-tune our model using COCO's training images. **Parameter setting.** We set $\sigma = 0.07, \{\tau_t\}_{t=1}^{T} = [0.8, 0.7.0.6, 0.5, 0.4], \phi = 5, \omega_f = 0.6, \omega_s = 0.4, K = 512$. The number of semantic queries is set to 50 and the number of instance queries is set to 150. We use a local crop scale between 0.05 and 0.4 for multi-crop augmentation. The images are resized to 448 as input. **Optimization Setting.** We train the model using adamw optimizer with a batch size of 16. The learning rate is linearly ramped up for the first 10k iterations to 0.000625. A cosine schedule is used to decay the learing rate to zero. It is noteworthy that our model is only trained for 160k iterations, while other models are trained for another multi-stage self-training. We use a cosine momentum schedule from 0.996 to 1 during training.

## 5.2 DATASETS AND METRICS

**Test Dataset.** We test unsupervised instance segmentation on COCO `test`val(Lin et al. (2014)), PASCAL VOS `val`(Everingham et al. (2010)) and UVO `val`(Wang et al. (2021)). We test unsupervised semantic segmentation on the COCOStuff-27(Caesar et al. (2018)) dataset. Following U2Seg (Niu et al. (2024)), we test unsupervised panoptic segmentation on Cityscapes `val`(Cordts et al. (2016) and COCO `val`. **Test Metrics.** We use the same evaluation protocol with U2Seg (Niu et al. (2024)). We use AP, $AP_{50}$, $AR_{100}$ to evaluate the unsupervised instance segmentation. We use Pixel Accuracy and mIoU to evaluate the unsupervised semantic segmentation. The clustering labels are mapped using Hungarian matching to class labels in the dataset. **Pretraining Dataset.** We use the ImageNet-1k (1.3M images) dataset for pretraining. Following U2Seg ( Niu et al. (2024)) on non-zero-shot evaluation, we also train our model over the combination of COCO and ImageNet images for 90k iterations.

## 5.3 EXPERIMENT RESULTS

**Self-supervised Instance Segmentation.** As shown in Table.2 and Table.3, our framework significantly outperforms other state-of-the-art methods on COCO, PASCAL VOC, and UVO. For class-agnostic unsupervised instance segmentation, our model achieves an increase of **+5.9** in $AP_{50}^{box}$, which is 25.8% of increase compared to U2Seg (Niu et al. (2024)). Our online pretraining framework can converge faster and achieve significant improvements over previous multi-stage alternating methods. Most importantly, for our method, the performance of the class-aware instance segmentation is higher of that in class-agnostic instance segmentation. However, this is reversed for U2Seg. This is because U2Seg adopts the multi-stage alternating strategy, where the pseudo-labels are regenerated at each round and thereby the classification learning is not stable and can not achieve better performance. This shows that the online clustering mechanism can better help model to capture the semantics information in the pretraining dataset.

**Self-supervised Semantic Segmentation.** As shown in Table.2, our framework also significantly outperforms other state-of-the-art methods on COCOStuff-27 for unsupervised semantic segmentation. Specifically, our model achieves an increase of **+5.9** in PixelAcc, which is 19.5% of increase compared to U2Seg and 20.9% of increase compared to STEGO.

**Self-supervised Panoptic Segmentation.** As shown in Table.2 and Table.3, our framework also significantly outperforms other state-of-the-art methods over panoptic segmentation on COCO and Cityscapes. For the zero-shot setting (solely trained on ImageNet), our method achieves an increase of **+6.8** in SQ on Cityscapes, which is 14.5% of increase compared to U2Seg, and achieves an increase of **+5.1** in PQ on COCO, which is 45.9% of increase compared to U2Seg. For the non-zero-shot setting (trained on combination of ImageNet and COCO), our method achieves an increase of **+8.0** in RQ on Cityscapes, which is 36.8% of increase compared to U2Seg, and achieves an increase of **+9.5** in SQ on COCO, which is 13.3% of increase compared to U2Seg.

## 5.4 ABLATION STUDIES

We identify three main hyperparameters in the design of our model for ablation studies, which are the Spatial and Feature similarity weights ($\omega_f, \omega_s$), the thresholds of each OAP layer $\{\tau_t\}_{t=1}^T$, and the number of online clusters (output dimension of the projection head). We evaluate all ablations on the unsupervised class-aware instance segmentation on UVO`val` dataset.

Table 4: Ablation studies for the Spatial and Feature similarity weights, the thresholds of each OAP layer, and the number of online clusters.

| $\omega_s, \omega_f$ | $AP^{mask}$ | $AR_{100}^{mask}$ | $\{\tau_t\}_{t=1}^T$ | $AP^{mask}$ | $AR_{100}^{mask}$ | $K$ | $AP^{mask}$ | $AR_{100}^{mask}$ |
|---|---|---|---|---|---|---|---|---|
| 0.6, 0.4 | 7.4 | 23.6 | 0.9-0.1 | 8.7 | 24.2 | 128 | 4.2 | 18.2 |
| 0.5, 0.5 | 7.6 | 24.1 | 0.5-0.1 | 4.5 | 18.6 | 512 | 8.3 | 24.3 |
| 0.4, 0.6 | **8.3** | **24.3** | 0.8-0.4 | 8.3 | **24.3** | 1024 | 8.8 | 25.4 |
| 0.0, 1.0 | 5.9 | 20.3 | | | | | | |

**Feature and Spatial similarity weights.** As shown in Table. 4, without considering the spatial similarity, i.e. setting $\omega_s = 0$, the performance drops significantly. By setting $\omega_s = 0.4$, our model achieves an increase of **+2.4** in AP$^{\text{mask}}$. This validates our design choices as illustrated in Section.4.2 The best performance is achieved when $\omega_s = 0.4, \omega_f = 0.5$. **Number of online clusters.** As shown in Table. 4, by using more clusters, our model can learn a finer representation granularity. This is also validated in U2Seg and previous self-supervised representation models. However, unlike DINO where a large number of 65536 are used, our model is designed for *dense* segmentation, a large number of clusters will throw out-of-memory and also slow down the training process. **Time-varied Layer Thresholds.** As shown in Table. 4, when we use a set of much lower thresholds, the performance drops significantly. This is because that a lower threshold (0.5) will make almost every edge to be coarsen. Instead, by using a more fine-grained set of thresholds (0.9,0.8,...,0.1), the OAP layer can identify more fine-grained groups of different semantic hierarchies. However, this would cost much time since more OAP layers are used. Instead, by setting an intermediate set of thresholds (0.8,...,0.4), our model can have comparable performance, while, more importantly, also cost less time.

# 6    SUMMARY

In this paper, we propose an efficient pseudo-mask generation algorithm, Online Agglomerative Pooling (OAP), to generate both the semantic-level and instance-level masks of one image within tens of milliseconds. Based on OAP, we propose the first online framework for self-supervised universal segmentation. A teacher-student framework is used, where we propose a simple but effective pretext task, Query-wise Self-distillation, specifically designed for self-supervised segmentation models. Our pretrained model achives state-of-the-art performance on unsupervised zero-shot instance segmentation, semantic segmentation, and panoptic segmentation tasks.

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
