# OpenReview forum: "Online Agglomerative Pooling for Scalable Self-Supervised Universal Segmentation"
_ICLR.cc/2025/Conference — ICLR 2025 Conference Withdrawn Submission_

### Official Review · Reviewer_FkbA · 2024-10-26

**Soundness:** 2
**Presentation:** 1
**Contribution:** 3
**Rating:** 5
**Confidence:** 3

**Summary:**

This paper introduces the Online Agglomerative Pooling (OAP) method for Unsupervised Universal Image Segmentation. The approach incorporates both an instance pooling layer and a semantic pooling layer to generate proposal masks following an "Identify then Merge" idea. The proposed OAP method can be seamlessly integrated into a self-training framework. Experiments are conducted on the COCO, VOC, UVO, and Cityscapes datasets to evaluate its effectiveness.

**Strengths:**

1. The method introduced in this work allows the integration of pseudo-mask generation into the self-training pipeline, streamlining the unsupervised segmentation training process
2. The Online Agglomerative Pooling (OAP) module is turned out to be efficient.
3. The proposed framework is shown to be effective and largely surpasses previous works.

**Weaknesses:**

1. The presentation of this work needs to be improved. Specifically, the details of the Online Agglomerative Pooling method are not thoroughly explained. Although the overall design of instance pooling and semantic pooling is understandable, a more in-depth explanation of the process is required. For instance, further clarification on graph initialization is needed, including how $V^0$ and $E^0$ are initialized and why $M^0$ has different size as $M_i$.
2. The effectiveness of the query-wise self-distillation approach lacks experimental validation. An ablation experiment is needed.
3. It would be better to have some visual comparisons.

**Questions:**

See Weaknesses part for questions.

---

### Official Review · Reviewer_HbdU · 2024-11-01

**Soundness:** 2
**Presentation:** 1
**Contribution:** 2
**Rating:** 3
**Confidence:** 1

**Summary:**

The paper works on self-supervised universal image segmentation. Unlike current methods that generate pseudo labels offline, this paper proposes to use the teacher-student online framework to online generate the pseudo masks. For this purpose, it proposes Online Agglomerative Pooling (OAP) to filter teacher network's output and uses so called query-wise self-distillation to match and align the outputs of teacher head and student heads. The experiments are conducted on some datasets and there are also some ablation study results for the thresholds, weights.

**Strengths:**

The goal of the paper is clear and the self-supervised learning for image segmentation is an important topic and task considering the annotation is expensive.

The idea is ok. There is no paper working on the self-supervised segmentation with the teacher-student framework as far as I know, although it is a very old method in image classification and object detection for semi-supervised learning.

**Weaknesses:**

The big problem is the presentation of the paper. It is very difficult to follow and some parts are hard to understand.

The experiments are limited. A more comprehensive evaluation is needed to trust the effectiveness of the proposed method.

**Questions:**

Line 74, what is “graph-partitioning”, it might be better to give a brief expression.

Line 75, what does “has better zero-shot performance for initialized training” mean? Better than what? What does “initialized training” mean? If the pseudo mask generation is of augmenting training data, why do we need online? For teacher-student style framework? But anyway, needs more clarification here. Why “makes the training unstable when switching to another round”? Shouldn’t the model has been converged in the last checkpoint? Otherwise, the training has not been finished? I think here the paper should explain why the online is more important.

Line 80, what does “online pretraining” mean?

Line 83, for “Our key insight is that semantically-similar patches are spatially close in group.”, any image or visualization examples to illustrate this?

Line 90, typo “suerpervised”.

Line 194, what does “level-specific” mean?

What is “local view”? Some augmented version of original images? What is “global view”? Is it the same as original view? Needs to clarify.

Line 252, if “edges are formed solely between nodes that are directly adjacent horizontally and vertically”, all tokens are potentially connected and could be a subtree for a certain token, no?

Eqn (3) is hard to understand.

Eqn (4), why w_s (S^s_ij)? But not  w_s S^s_ij? How to ensure S^f_ij and S^s_ij at the same scale, if using w_f+w_s=1 to linearly combine them. Why S_ij in [-1,1]? How to find the threshold?

Why averaging features of tokens inside its mask can cause the bad merging?

Line 311, why “each mask corresponds to a single connected region, which is predominantly suitable for instance segmentation tasks. However, semantic segmentation often requires masks that encompass multiple disjoint regions belonging to the same semantic class.”. The semantic regions could also be small but instance segmentation region could be large. Why this is a problem.

The selection of thresholds is too heuristic.

What is T, sometimes it is the transpose, sometimes it is layer or hierarchy? Or iteration number. Too vague.

Why zero-initialize the adapter parameters can make OAP generate meaningful masks during the early stages of training (Line 359)?

Line 361, the mask decoder must output both instance-level masks and semantic-level mask?

Can’t understand how to match the output from student head to the teacher head.

What are the instance pooling and semantic pooling?

Why OAP’s output can can be used to train the student network. Is that because the original input to the teacher network or OAP can keep high-quality masks. This is unclear.

It is recommended to put all tables and figures to the top of each page.

Line 434, textttval?

Table 2, why use AP^{box} to evaluate results of instance segmentation. Shouldn’t use mask AP or something like that?

Any comparison to UnSAM?

In the table 2 of CutLER paper, it has a very comprehensive evaluation. The proposed method should do an apple-to-apple comparison on the wide benchmarks.

Any qualitative comparison on generated pseudo masks with other methods?

---

### Official Review · Reviewer_9eoc · 2024-11-03

**Soundness:** 3
**Presentation:** 3
**Contribution:** 3
**Rating:** 6
**Confidence:** 4

**Summary:**

This paper proposes a novel approach called Online Agglomerative Pooling (OAP) aimed at improving self-supervised image segmentation by addressing the limitations of multi-stage, offline pretraining methods that rely on iterative pseudo-mask generation. The key objective is to enable efficient, high-quality pseudo-mask creation during each training step, eliminating offline steps, which results in faster convergence and enhanced performance. The framework leverages a teacher-student architecture with Query-wise Self-Distillation, ensuring that OAP generates segmentation masks efficiently and scales well to large datasets. Experimental results demonstrate that OAP achieves state-of-the-art performance across various segmentation tasks, including zero-shot instance, semantic, and panoptic segmentation.

**Strengths:**

The proposed approach enhances existing methods effectively for its intended purpose. By leveraging U2Seg's task foundation, it addresses the complexity of prior offline or alternative iterative processes through the simplification of an online approach using Online Agglomerative Pooling (OAP). This innovation integrates elements from notable frameworks like Mask2Former’s mask decoder and DINO’s local-to-global matching, embedding them into the context of universal segmentation tasks.

The introduction of OAP successfully identifies and resolves step-by-step limitations inherent in prior models, explicitly clarifying the boundaries of each stage and providing solutions for these challenges. The Identify step within OAP is uniquely enhanced to incorporate the evolving nature of node features across layers by integrating spatial similarity to maintain consistency during transitions to the merge step. The merge step itself is refined by introducing a normalization mechanism that ensures semantically different yet spatially close nodes are initially merged, with additional emphasis on suppressing boundary interference.

For semantic pooling, the challenge of segmenting disjoint regions into a single semantic class—an inherent issue in semantic segmentation as opposed to instance segmentation—is adeptly tackled by constructing a fully connected graph from the output of instance pooling. This allows multiple disjoint regions to be appropriately recognized and combined under the same semantic label.

Moreover, the method employs a decreasing threshold strategy, which considers time-variant tendencies. This approach ensures that earlier pooling layers handle high-level background mergers while later layers consolidate more complex foreground semantics. Such a design choice optimizes the model's ability to produce hierarchical and detailed segmentations while maintaining efficiency.

**Weaknesses:**

The paper introduces Online Agglomerative Pooling (OAP) algorithm and Query-wise Self-distillation, which significantly improve the performance of self-supervised universal segmentation. However, there is a lack of qualitative examples illustrating the specific problems addressed by each method and the extent of improvement achieved. The experimental section could benefit from a more detailed description of the drawbacks of each method, how they were overcome, and the resulting performance gains. This would more effectively convey the superiority of the proposed techniques.

The authors emphasize the advantages of the single-stage approach compared to the multi-stage approach, but they do not provide a concrete explanation of the time-saving benefits. Quantitatively presenting the differences in time consumption between the multi-stage and single-stage approaches would more clearly demonstrate the efficiency of the single-stage approach.

The proposed method has the advantage of addressing the data annotation cost issue and approaching the problem without labels. However, it may have lower performance compared to semi-supervised or weakly supervised approaches, which is a limitation. Although significant performance improvements have been made compared to previous studies, the effectiveness of this approach may be questioned. Recently, there have been attempts to solve the annotation cost problem using stable diffusion or text-to-image generation models. Including a discussion on cost comparison and feasibility with these approaches would make the research more meaningful.

It would be beneficial to provide a detailed analysis of the resource usage of the SCC algorithm used in OAP. An in-depth discussion is needed on whether there are only advantages in terms of CPU usage and time consumption, if there are any drawbacks, and if there are any aspects that need improvement.

The datasets used in the experiments mainly consist of images featuring general objects. It is necessary to verify if similar effects can be observed on datasets from different domains. It would be helpful if the authors present their views on whether the proposed technique can be extended to a wider range of applications.

**Questions:**

I think more detailed ablation study for the proposed method would be beneficial. In this sense, Figure 4 is good for understanding and showing the effectiveness of the method.

Please discuss the usefulness of the self-supervised approach compared to semi-supervised or, weakly supervised method.

P9 L434 COCO textttval -> COCO \texttt{val} ?

---

### Author Response · Authors · 2024-11-14

We would like to express our sincere gratitude to the ACs, Reviewers, and PCs for their valuable time and thoughtful feedback on our paper. After careful consideration, we have decided to withdraw the paper, but we truly appreciate the effort and constructive comments that have significantly contributed to the improvement of our work. Thank you once again for your support and assistance.

---

### Note · Authors · 2024-11-14

I have read and agree with the venue's withdrawal policy on behalf of myself and my co-authors.